# Neurotherapeutics for Attention Deficit/Hyperactivity Disorder (ADHD): A Review

**DOI:** 10.3390/cells10082156

**Published:** 2021-08-21

**Authors:** Katya Rubia, Samuel Westwood, Pascal-M. Aggensteiner, Daniel Brandeis

**Affiliations:** 1Department of Child & Adolescent Psychiatry, Institute of Psychiatry, Psychology & Neurosciences, King’s College London, De Crespigny Park, London SE5 8AF, UK; s.westwood4@wlv.ac.uk; 2Department of Social Genetics and Developmental Psychiatry, Institute of Psychiatry, Psychology & Neurosciences, King’s College London, De Crespigny Park, London SE5 8AF, UK; 3Department of Child & Adolescent Psychiatry, Transcampus, Dresden University, 01307 Dresden, Germany; 4Department of Psychology, Wolverhampton University, Wolverhampton WV1 1LY, UK; 5Department of Child and Adolescent Psychiatry and Psychotherapy, Medical Faculty Mannheim, Central Institute of Mental Health, Heidelberg University, 68159 Mannheim, Germany; Pascal.Aggensteiner@zi-mannheim.de (P.-M.A.); daniel.brandeis@zi-mannheim.de (D.B.); 6Department of Child and Adolescent Psychiatry and Psychotherapy, Hospital of Psychiatry, Psychiatric Hospital University, University of Zürich, 8032 Zürich, Switzerland; 7Neuroscience Center Zürich, Swiss Federal Institute of Technology and University of Zürich, 8057 Zürich, Switzerland

**Keywords:** attention deficit hyperactivity disorder (ADHD), functional magnetic resonance imaging (fMRI), neurofeedback, EEG-neurofeedback, fMRI-neurofeedback, brain stimulation, transcranial magnetic stimulation (TMS), transcranial direct current stimulation (tDCS), trigeminal nerve stimulation (TNS)

## Abstract

This review focuses on the evidence for neurotherapeutics for attention deficit/hyperactivity disorder (ADHD). EEG-neurofeedback has been tested for about 45 years, with the latest meta-analyses of randomised controlled trials (RCT) showing small/medium effects compared to non-active controls only. Three small studies piloted neurofeedback of frontal activations in ADHD using functional magnetic resonance imaging or near-infrared spectroscopy, finding no superior effects over control conditions. Brain stimulation has been applied to ADHD using mostly repetitive transcranial magnetic and direct current stimulation (rTMS/tDCS). rTMS has shown mostly negative findings on improving cognition or symptoms. Meta-analyses of tDCS studies targeting mostly the dorsolateral prefrontal cortex show small effects on cognitive improvements with only two out of three studies showing clinical improvements. Trigeminal nerve stimulation has been shown to improve ADHD symptoms with medium effect in one RCT. Modern neurotherapeutics are attractive due to their relative safety and potential neuroplastic effects. However, they need to be thoroughly tested for clinical and cognitive efficacy across settings and beyond core symptoms and for their potential for individualised treatment.

## 1. Introduction

Attention-deficit/hyperactivity disorder (ADHD) is characterised by persisting and impairing symptoms of age-inappropriate inattention and/or hyperactivity/impulsivity (DSM-5) (American Psychiatric Association, 2000) [1]. It is one of the most common childhood disorders with a worldwide prevalence of around 7% (Thomas et al., 2015) [2]. Problems persist into adulthood in a substantial proportion of cases, and they are associated with comorbidities and poor academic and social outcomes (Thomas et al., 2015) [2].

ADHD patients have deficits in higher-level cognitive functions necessary for mature adult goal-directed behaviours, in so-called “executive functions” (EF), that are mediated by late developing fronto-striato-parietal and fronto-cerebellar networks (Rubia, 2013) [3]. The most consistent deficits are in “cool” EF such as motor response inhibition, working memory, sustained attention, response variability and cognitive switching (Pievsky & McGrath, 2018; Rubia, 2011; Willcutt et al., 2008) [4,5,6], as well as in temporal processing, in particular in time discrimination and estimation tasks (Noreika et al., 2013; Rubia et al., 2009) [7,8]. However, impairment has also been found in so-called “hot” EF functions of motivation control and reward-related decision making, as measured in temporal discounting and gambling tasks. However, evidence for hot EF deficit has been more inconsistent than for cool EF [5,8,9] (Noreika et al., 2013; Plichta & Scheres, 2014; Willcutt et al., 2008), in line with the diagnostic criteria. Evidence for cognitive deficits is more consistent in children than adolescents or adults with ADHD [6,10] (Groen et al., 2013; Pievsky & McGrath, 2018). There is furthermore considerable heterogeneity in cognitive impairments, with over 30% of patients showing no EF impairments (Nigg et al., 2005; Roberts et al., 2017) [11,12].

The most successful treatment is with psychostimulant medication which enhances catecholamines in the brain, reaching an effect size of ~0.8, with about 70% of patients with ADHD responding to it [13] (Cortese et al., 2018). Functional magnetic resonance imaging (fMRI) studies have shown that stimulant medication increases the activation of inferior frontal and striatal regions and their interconnectivity and decreases activation in areas of the default mode network [14] (Rubia et al., 2014), both of which are likely responsible for improvements in cognitive functioning [15,16] (Coghill et al., 2014; Pievsky & McGrath, 2018). Second-line treatment is with noradrenaline transporter/receptor blockers, Atomoxetine and Guanfacine, which also enhance brain catecholamines with effect sizes of 0.56 and 0.67, respectively [13] (Cortese et al., 2018). Stimulant prescription has increased dramatically over the last decades worldwide, which is controversial due to abuse and diversion potential. Furthermore, stimulants commonly have adverse effects on sleep and appetite as well as causing irritability, nausea/vomiting, abdominal pain, headaches, labile mood and growth suppression, although they are typically non-serious and can be transient [13] (Cortese et al., 2018). Moreover, only 50% of patients tolerate it sufficiently, caution is indicated for certain comorbid conditions (such as cardiovascular malfunctions and sleep problems) and adherence can be poor, in particular in adolescence. Importantly, longer-term efficacy has not been demonstrated in meta-analyses, nor in observational or epidemiological studies [13,17] (Cortese et al., 2018; Swanson, 2019), although there is controversy (Coghill, 2019) [18].

While the efficacy of stimulant medication for treating ADHD was a chance finding, as it was originally used for other medical conditions such as bronchodilatation, headache, and blood pressure [19] (Connolly et al., 2015) and the first neurofeedback treatment in ADHD also used EEG conditioning developed for seizure control [20] (Lubar & Shouse, 1976), modern neurotherapeutics have the advantage that they can directly target the key brain function deficits that have been found in ADHD over the past decades. There has been substantial research on brain function differences in ADHD relative to healthy controls with electroencephalography (EEG) since the 1970s (e.g., Satterfield, 1973; Satterfield et al., 1973) [21,22] and with fMRI over the past 2.5 decades that have provided us with neurofunctional biomarkers that could be targeted with neurotherapeutics, such as neurofeedback or non-invasive brain stimulation techniques.

## 2. Search Methods for the Review

For this review, Web of Knowledge, Scopus, PubMed, Ovid, Google Scholar, psyarxiv and bioRxiv (up until the end of May 2021) were searched with the following key words: ‘electroencephalography’, ‘EEG’, ‘event-related potentials’, ‘ERP’, ‘theta-beta ratio’, ‘TBR’, ‘functional magnetic resonance imaging’, ‘fMRI’, ‘neurofeedback’, ‘EEG-neurofeedback’, ‘electrophysiology-neurofeedback’, ‘fMRI-neurofeedback’, ‘functional magnetic resonance imaging-neurofeedback’, ‘NIRS-Neurofeedback’, ‘near-infrared spectroscopy-neurofeedback’, ‘non-invasive brain stimulation’, ‘transcranial electric stimulation’, ‘transcranial direct current stimulation’, ‘tDCS’, ‘transcranial magnetic stimulation’, ‘repetitive transcranial magnetic stimulation’, ‘rTMS’, ‘transcranial electric stimulation’, each in combination with ‘attention deficit/hyperactivity disorder’, ‘ADHD’, ‘hyperkinetic disorder’, ‘inattention’, ‘hyperactivity’, ‘impulsivity’ and/or ‘meta-analysis’. In addition, the reference lists of articles and reviews or meta-analyses were also hand-searched. Single case studies were excluded.

## 3. Functional Neuroimaging Markers of ADHD That Could Provide Targets for Neurotherapeutics

### 3.1. Electrophysiological Biomarkers

Electrophysiology findings in ADHD showed that increased slower oscillations such as delta, theta or alpha during resting conditions, but also faster beta frequencies bands, are most relevant to ADHD (Loo & Makeig, 2012) [23]. The oscillatory or spectral profile reflects maturation and arousal problems, since particularly slower frequencies decrease with age. An increasingly controversial finding in ADHD is a higher frontocentral theta/beta ratio (TBR) [24] (Snyder et al., 2015) which has been related to reduced attention, hypoarousal or maturational lag, suggesting a robust link between ADHD and resting EEG markers. During the last decades, scientific efforts to replicate this hypothesis did not show consistent TBR increase in ADHD despite maturational effects [25,26,27,28] (Buyck & Wiersema, 2014, 2015; Liechti et al., 2013), and a relation between TBR and arousal has been questioned [29] (Clarke et al., 2019). A meta-analysis about TBR in ADHD showed that the TBR effect size is negatively related to the year of publication and might be related to methodological factors and to a trend for increasing TBR over the years in healthy controls, which may be related to decreased sleep duration, diminishing differences to ADHD [30] (Arns et al., 2012). Importantly, advances in the field showed that the heterogeneity within ADHD might explain the inconsistent findings. Indeed, it was shown that subgroups of patients with ADHD have increased TBR [31] (Clarke et al., 2011), but only in three of the five EEG clusters, with 60% of children with ADHD showing increased theta activity. A more recent study showed that high TBR is present in 35% of the ADHD population [32] (Bussalb et al., 2019). However, the concept of TBR as a biomarker for ADHD could potentially be confounded by differences in concentration, cognitive effort, activation, and drowsiness [33] (Drechsler et al., 2020), consistent with findings that theta activity increases in ADHD appear only after longer EEG recordings [34] (Zhang et al., 2019). Further, a recent review on resting EEG power research in ADHD concluded that given the current evidence in the field, it would be premature to make definitive statements about the utility of the TBR ratio as a diagnostic test for ADHD [35] (Clarke et al., 2020). Importantly, recent EEG-NF studies which assume deviating TBR have taken this into account, proposing a cut-off for TBR-NF (i.e., >4.1) and thus applying TBR-NF only to the subgroup with high TBR ratio (Arnold et al., 2020; Bioulac et al., 2019) [36,37].

Compared to this controversial research regarding inconsistently altered electrophysiological oscillations, there are somewhat more consistent findings concerning event-related potentials (ERPs). ERPs are defined as a task-locked activity, reflecting cognitive, sensory or motor brain responses. Different ERP components showed deviations in ADHD for stimulus discrimination, resource allocation, inhibition, preparation, error detection, and conflict processing [38,39] (Barry et al., 2003, Johnstone et al., 2013). However, these alterations seem to be non-specific to ADHD and provide only limited relevance as diagnostic biomarkers [23] (Loo & Makeig, 2012). A current meta-analysis [40] (Kaiser et al., 2020) found significant and moderate to large effects for specific ERPs associated with late cognitive processing related to attentional preparation and resource allocation, such as P300 and contingent negative variation (CNV); however, the results were characterized by substantial heterogeneity and modest effect sizes which limit their use for clinical applications. Importantly, there is a need to systematically investigate these components, since most of the studies used different tests and measures, making reliable interpretation regarding classification accuracy and effect size particularly difficult [41] (Gamma & Kara, 2020).

### 3.2. fMRI Biomarkers

The past two decades of MRI research have consistently shown evidence for underlying brain structure and function deficits in ADHD. Therefore, ADHD is now considered a neurodevelopmental disorder. Meta- and mega-analyses of structural volumetric studies in ADHD have shown reduced gray matter in subcortical regions, most prominently the basal ganglia and insula [42,43,44,45] (Hoogman et al., 2017; Lukito et al., 2020; Nakao et al., 2011; Norman et al., 2016), but also limbic areas such as the amygdala and hippocampus [42] (Hoogman et al., 2017) and reduced gray matter, surface area and cortical thickness in frontal, temporal and parietal regions [44,45,46] (Hoogman et al., 2019; Lukito et al., 2020; Norman et al., 2016). Furthermore, there is evidence for a delay in the peak of cortical thickness and surface area maturation in frontal, temporal and parietal regions (Shaw et al., 2007; Shaw et al., 2012) [47,48]. White matter tracts have also been found to be impaired in the disorder, most prominently fronto-striatal, fronto-cerebellar, interhemispheric tracts and long-distance tracts such as fronto-occipital tracts (Aoki et al., 2018; Chen et al., 2016) [49,50] (for review, see Rubia, 2018) [51].

fMRI studies have provided consistent neurofunctional biomarkers in ADHD, several of which have been targeted with neurotherapeutics. ADHD has been associated with relatively widespread dysfunctions, mostly underactivations relative to healthy controls, involving the lateral inferior and dorsolateral prefrontal cortex and its connections to the basal ganglia, as well as medial frontal, cingulate and orbital frontal regions and the dissociated fronto-parietal, fronto-limbic and fronto-cerebellar networks that they form a part of (Rubia, 2018) [51].

Several fMRI meta-analyses have been published on ADHD, the majority including fMRI studies using cool EF tasks. They show cognitive domain-dissociated brain dysfunctions in several fronto-striatal, fronto-parietal and fronto-cerebellar regions in ADHD. A well-replicated finding across our three meta-analyses of whole-brain fMRI studies of cognitive and motor inhibition, the latest and largest including 1001 ADHD patients, is that people with ADHD relative to healthy controls have consistently reduced activation in key regions of cognitive control: in the right inferior prefrontal cortex (IFC)/anterior insula, the supplementary motor area (SMA), anterior cingulate cortex (ACC) and striatal regions [44,45,52] (Hart et al., 2013; Lukito et al., 2020; Norman et al., 2016). Similar findings were observed in smaller meta-analyses focusing on inhibition tasks, some including left IFC [53,54,55] (Cortese et al., 2012; Lei et al., 2015; McCarthy et al., 2014) and others also finding DLPFC underactivation [53,54,55] (Cortese et al., 2012; Lei et al., 2015; McCarthy et al., 2014). Our meta-analysis of a relatively wide range of fMRI studies of attention tasks, including selective, divided and sustained attention, as well as alerting and mental rotation, found reduced activation in 171 ADHD patients relative to 178 healthy controls in the right hemispheric dorsal attention network, comprising right DLPFC, right inferior parietal cortex and caudal parts of the basal ganglia and thalamus. In addition, ADHD patients had increased activation relative to controls in right cerebellum and left cuneus, presumably compensating for the reduced activation of the frontal part of the dorsal DLPFC-parieto-cerebellar attention network [52] (Hart et al., 2013). Another meta-analysis reported significantly reduced activation in right anterior cingulate during attention tasks from a sub-analysis of 11 fMRI datasets [55] (Cortese et al., 2012). A meta-analysis of fMRI studies of timing functions, including 11 fMRI studies of time discrimination, time estimation, motor timing and temporal discounting (temporal foresight), showed consistently reduced activation in 150 ADHD patients relative to 145 healthy controls in left IFC, left inferior parietal lobe and right lateral cerebellum [56] (Hart et al., 2012), all key regions mediating timing functions [57] (Wiener et al., 2010). A meta-analysis of n-back working memory (WM) fMRI studies showed that 111 ADHD patients relative to 113 controls had reduced activation in bilateral middle and superior PFC and left MFC/ACC [53] (McCarthy et al., 2014), although some large-numbered studies and other meta-analyses also found right and left IFC underactivation [55,58] (Cortese et al., 2012; van Ewijk et al., 2015). The right IFC dysfunction during cognitive control tasks, in particular, has been shown to be disorder-specific to ADHD relative to OCD and to ASD in two large comparative meta-analyses [44,45] (Lukito et al., 2020; Norman et al., 2016). The findings of domain-dissociated deficits in distinct IFC/ACC/SMA fronto-striato-thalamic (inhibition), right DLPFC fronto-striato-thalamo-parietal (for attention), bilateral DLPFC, IFC and MFC/ACC (working memory) and left IFC-parieto-cerebellar regions (timing) in ADHD patients for these different cognitive domains suggest that ADHD patients suffer from multisystem deficits compromising different fronto-striato-parieto-cerebellar networks that mediate several cognitive domains [51] (Rubia, 2018).

In addition to deficits in several of these lateral fronto-striato-parietal and fronto-cerebellar regions that mediate so-called “cool” EF, children with ADHD have also shown reduced activation in the ventromedial prefrontal cortex (vmPFC), orbitofrontal cortex (OFC) and striato-limbic regions during tasks of “hot” EF such as reward-related decision making or temporal discounting tasks. However, deficit findings have been less consistent (Plichta & Scheres, 2014; Rubia, 2018) [9,51].

There is furthermore evidence for reduced inter-regional functional connectivity between these task-relevant regions during cognitive tasks and during the resting state, in particular in the dorsal and ventral attention and cognitive control networks (Rubia, 2018; Sripada et al., 2014; Sripada et al., 2014). [51,59,60].

However, not only task-positive regions have been found to be abnormal in function in ADHD, but also areas of the default mode network (DMN), which comprise ventromedial frontal cortex, posterior cingulate, precuneus and inferior parietal and temporal regions and which is thought to reflect task-irrelevant thoughts and mind wandering [61] (Raichle, 2015). Thus, several of the above-reviewed meta-analyses and individual fMRI studies also report increased activation in ADHD patients in regions of the DMN such as in rostromedial prefrontal cortex during interference inhibition [52] (Hart et al., 2013), posterior cingulate and precuneus during motor inhibition, sustained attention and other cognitive control [52,62,63] (Hart et al., 2013, Christakou et al., 2013; Salavert et al., 2018) and timing tasks [56] (Hart et al., 2012). The findings suggest that ADHD patients have less control over their interoceptive attention orientation and mind-wandering [64] (Bozhilova et al., 2018), which intrudes into their already weak exteroceptive attention processes, likely causing enhanced inattention and impulsiveness. This immature pattern of poor activation of task-relevant and age-correlated task-positive brain activation networks and of decreased deactivation of the DMN are likely underlying the poor performance in ADHD on attention-demanding higher-level cognitive control tasks (Rubia, 2018) [51].

The most consistently found dysfunctional regions, in particular right IFC followed by right DLPFC, ACC, right inferior parietal lobe or the basal ganglia, could potentially be used as targets for neurotherapeutics. Some of these regions such as IFC, DLPFC and ACC have already been used as targets of neuromodulation in fMRI/NIRS-neurofeedback or for brain stimulation therapies. Furthermore, with fMRI-NF, entire networks that are affected in ADHD could also potentially be targeted, such as dorsal and ventral attention or the cognitive control systems [59] (Sripada et al., 2014). Downregulating the DMN could potentially also be a suitable, yet unexplored neurotherapeutic target for fMRI-NF. Given evidence for the anti-correlation between the IFC/DLPFC and the DMN [59] (Sripada et al., 2014), the upregulation of IFC/DLPFC with brain stimulation or neurofeedback may indirectly downregulate areas of the DMN, which we have indeed shown to be the case in ADHD patients after fMRI-NF of right IFC (Rubia et al., 2019) [65].

## 4. Neurotherapeutics in ADHD

One of the most revolutionary findings of the last decade of neuroimaging has been the discovery of high brain plasticity, in particular in childhood/adolescence when the brain is still developing [66,67] (Jancke, 2009; Rapoport & Gogtay, 2008), but also in mid and older adulthood [68,69] (Draganski et al., 2004; Draganski & May, 2008). Even a few weeks or months of training of a particular skill in mid and older adults, for example, juggling [68,69] (Draganski et al., 2004; Draganski & May, 2008), learning for an exam [70] (Draganski et al., 2006) or learning to meditate [71] (Dodich et al., 2019) can change the structure of the brain. These insights into the brain’s neuroplastic potential make novel neuromodulation treatments, such as non-invasive brain stimulation or neurofeedback, attractive clinical interventions [51,72] (Rubia, 2018, Ashkan et al., 2013). This is even more relevant to early stages of disorders in young people where it is likely to be most effective [73] (Anderson et al., 2011), with evidence showing that children and adolescents show accelerated neural plasticity compared to adults after brain stimulation [74] (Brunoni et al., 2012).

The establishment of neurofunctional biomarkers for ADHD with EEG and fMRI studies over the past decades has made it possible to target these biomarkers using neurotherapeutics. Given the evidence for electrophysiological and neuroimaging functional deficits in ADHD, it seems plausible that treatments that try to reverse these underlying brain function deficits could potentially be promising, given that they are targeting the key neurobiological abnormalities associated with the disorder. EEG-NF has already been applied to ADHD for over 45 years with relatively inconsistent findings. fMRI or NIRS-neurofeedback is still very much in its infancy with too few and underpowered applications to provide a clear insight on potential efficacy. There has been an exponentially increasing number of non-invasive brain stimulation studies over the past 10 years. Studies have, however, been relatively small numbered with very heterogenous study designs. Consequently, findings have been inconsistent with respect to improving cognition, with very little evidence so far on improving clinical behaviour. The following sections will review these clinical applications of neuromodulation in populations with ADHD.

### 4.1. Neurofeedback

Neurofeedback (NF) is an operant conditioning procedure that, by trial and error, teaches participants to volitionally self-regulate specific regions or networks through real-time audio or visual feedback of their brain activation, which can be represented on a PC. For children, this can be gamified in an attractive way. Given that ADHD is typified by poor self-control [75] (Schachar et al., 1993), and enhancing brain-self-control is the target of NF, ADHD is the psychiatric disorder where NF has been most applied, using electrophysiological neurofeedback (EEG-NF).

#### 4.1.1. EEG-NF

EEG-NF trains self-regulation of oscillatory or task-related EEG-markers associated with ADHD, such as increased theta and TBR linked to compromised activation; decreased sensorimotor rhythm (SMR) related to impaired state regulation and sleep; and attenuated task-related slow cortical potentials (SCP) such as the CNV correlated with impaired preparation and activation (standard protocols) [76] (Arns et al., 2014).

Despite the fact that EEG-NF has been used to treat ADHD for 45 years, the majority of the studies had important methodological shortcomings like the lack of an appropriate control condition, randomisation, unblinded outcome measures, non-standardized feedback methods, limited or no reporting of self-regulation and appropriate learning. During the last two decades, large improvements have been made to address these major drawbacks, resulting, for example, in a very recent consensus publication on the reporting and experimental design of neurofeedback studies [77] (Ros et al., 2020).

##### Meta-Analyses of EEG-NF

During the last decade, a large number of meta-analyses were published which scrutinize the clinical efficacy of EEG-NF in ADHD. The first meta-analysis based on ten controlled studies reported large effect sizes in favour of EEG-NF when parents rated the clinical outcome of inattention or for impulsivity measured in tests, as well as non-inferiority compared to the gold standard of stimulant medication treatment; it therefore recommended EEG-NF as “efficacious and specific”, which means that the treatment has been shown to be statistically superior to credible sham therapy in at least two independent studies [78] (Arns et al., 2009) (for updated, more stringent criteria, see Arns et al., 2020) [79].

More than ten years and more than ten meta-analyses later [76,78,80,81,82,83,84,85,86,87,88] (Arns et al., 2009, 2014; Cortese et al., 2016; Van Doren et al., 2019; Micoulaud-Franchi et al., 2014; Riesco-Matías et al., 2021; Sonuga-Barke et al., 2013; Yan, et al., 2019; Lambez et al., 2020; Bussalb et al., 2019; Hodgson et al., 2014), the latest comprehensive meta-analysis to date reported significant albeit small to medium effect sizes and inferiority compared to stimulants [83] (Riesco-Matías et al., 2021). This drop of more than half of the effect size (for a historical/chronological viewpoint, see Figure 1) is interesting and probably related to the growing research using stricter control conditions and improved scientific standards for EEG-NF studies, which will be discussed in the following.

The first meta-analysis [78] (Arns et al., 2009) included non-randomised studies which are considered a weak experimental design to determine clinical efficacy [89] (Norris & Atkins, 2005), whereas randomised controlled trials (RCT) are considered gold-standard in clinical research. The following meta-analysis [90] (Sonuga-Barke et al., 2013) addressed this issue by including only RCTs, together with the inclusion of blinding criteria of the clinical outcome, such as ADHD core symptoms. These authors introduced the term of “probably blinded” raters, which refers to the assessment, most often by teachers, who probably do not know to which treatment the patient was allocated. These two new requisites blunted the clinical effect which still remained significant for unblinded raters (such as parents) with medium effect sizes but was reduced to a trend-level for the probably blinded raters. Following these new insights, the recommendation to consider EEG-NF in ADHD as efficacious and specific was ameliorated.

One year later, Micoulaud-Franchi et al. [82] (Micoulaud-Franchi et al., 2014) updated Sonuga-Barke’s meta-analysis of 2013 [84] (Sonuga-Barke et al., 2013), including the subdomains of the core ADHD symptoms, i.e., inattention, hyperactivity and impulsivity. When evaluating the core symptom domains separately, a significant effect emerged also for the probably blinded raters, but only for the inattention subdomain.

Subsequently, two years later, an update of Sonuga-Barke’s meta-analysis was published by the same group [80] (Cortese et al., 2016) on behalf of the European ADHD guidelines group, incrementing the analysis from 8 to 13 RCTs with parent-ratings and from 4 to 8 RCTs with probably blinded ratings. This updated meta-analysis resulted in insignificant findings for all probably blinded ratings, including inattention, but still showed a significant medium effect size for parents’ ratings. The discrepancy regarding the blinded findings in the subdomain of inattention in Micoulaud-Franchi [82] (Micoulaud-Franchi et al., 2014) appears due to selecting different blinded outcomes in the same studies.

The meta-analysis of Cortese [80] (Cortese et al., 2016) also reported an exploratory sensitivity analysis including only three EEG-NF studies that used standard protocols [76] (Arns et al., 2014), where the effects on ADHD symptoms became also significant for probably blinded raters, but subsequent large standard NF trials [36,91] (i.e., Arnold et al., 2020; Strehl et al., 2017) could not substantiate this. Importantly, Bussalb et al. [32] (Bussalb et al., 2019) in their meta-analysis systematically evaluated further factors which influenced the efficacy of NF. They concluded that the intensity of NF but not the treatment duration was associated with higher efficacy, teachers were less sensitive to patients’ symptoms and suggested that NF needs to be evaluated with placebo-controlled interventions.

As can be observed from this, progress has been made to enhance the quality and certainty of the consideration and evaluation of the efficacy of EEG-NF in ADHD. Neurofeedback should be considered an umbrella term since there exist a large number of different training modalities that are only limited by the available technology (such as Coherence training, asymmetry feedback, etc). This issue is of paramount importance and a standardization should be aimed for. To date, the already mentioned standard protocols fulfil these criteria and so far, very recently, a few larger studies were published.

The latest comprehensive meta-analysis [83] (Riesco-Matías et al., 2021) addressed an additional important point, which is the selection of an adequate control group, and compared EEG-NF vs. non-active control groups (waiting-list controls, treatment as usual) and active control groups. The main findings showed superiority of EEG-NF compared to non-active control groups for parent ratings and for the inattention subdomain rated by probably blinded raters, resembling the findings of Micoulaud-Franchi et al. [82] (Micoulaud-Franchi et al., 2014). However, when EEG-NF was compared with an active control condition, such as pharmacotherapy, EEG-NF was no longer superior. These findings underline the importance of considering active elements in control conditions, and the need to grade these active elements consistently across Neurofeedback and other neurotherapies studies. The recent consensus statement on evidence-based ADHD treatments excluded studies and meta-analyses with non-active or heterogeneous controls such as waiting control or treatment as usual [92] (Faraone et al., 2021). However, this approach may underestimate some genuine NF-effects in real life settings that are also detectable by blinded raters or are slower to develop.

##### Other Aspects of EEG-NF

Still, it is also important to take into consideration the cost-benefit aspects and preferences for the individual patient. As discussed above, pharmacotherapy has limitations due to side effects and no consistent longer-term effects. One recent meta-analysis addressed the question of longer-lasting effects of EEG-NF six months after treatment and showed small to medium effects in favour of neurofeedback when compared to non-active conditions and comparable effects relative to active conditions, mainly pharmacotherapy, contrasting with the superiority of the active control conditions shortly after treatment [81] (Van Doren et al., 2019). EEG-NF thus seems to have a delayed beneficial effect, as for example in a study where the superiority of stimulants over NF observed at treatment end [93] (Geladé et al., 2016) was no longer significant at the six-month follow-up, and ADHD core symptoms compared to a semi-active (physical exercise) control condition were similar at treatment end but became reduced with NF relative to the exercise control condition at follow-up [93] (Geladé et al., 2016). However, contradictory findings from the largest study to date, which assessed longer-term effects of EEG-NF, showed that although the improvement of ADHD core symptoms relative to the baseline remained large and stable after treatment at six-month follow-up, it was no longer superior to a semi-active condition [94] (Aggensteiner et al., 2019), suggesting considerable unspecific long-term effect.

In general, the specificity of the efficacy of EEG-NF is controversial and still under debate. During the last decade, disentangling the true effect related to neuromodulation from non-specific effects has been under investigation. In Strehl et al. [91] (Strehl et al., 2017), this was addressed by comparing EEG-NF with a semi-active control EMG-BF group, controlling for unspecific effects, such as the high-tech training setting, interaction, learning, time, motivation, expectation and effort, which showed clinical superiority in favour of EEG-NF one month after treatment end. Controlling for these factors is highly important since the clinical effects of this kind of time-consuming training might otherwise be attributed to unspecific psychosocial [95] (Wood & Kober, 2018) or placebo effects which seem particularly strong with treatments that involve high-tech settings [96,97,98,99,100] (Thibault et al., 2016, 2017; Thibault et al., 2018; Thibault & Raz, 2016, Schönenberg et al., 2021). To control for these aspects, a sham-feedback condition is often considered a gold standard in intervention research.

The recent large double-blind placebo-controlled study of the Collaborative Neurofeedback Group [36] (2020) which compared TBR-NF with a double-blind sham-NF placebo group not only followed this approach, but also introduced individualisation by selecting only participants with an elevated TBR. The results showed large uncontrolled clinical effects until 13-month follow-up in both groups relative to baseline and a reduced need for medication in the neurofeedback group at follow-up but failed to demonstrate clinical superiority for EEG-NF despite more TBR learning in the NF than in the sham group (67% vs. 59%) (Arnold et al., 2020) [36]. The mechanism which explains the large nonspecific clinical effects in both groups remains unclear.

Given that the main aim of neuromodulation is to self-regulate the trained parameters, improvement of brain modulation should be related to clinical improvement and explain clinical outcome. This relation remains understudied [101] (Zuberer et al., 2015) and is complicated due to delayed effects, as discussed above, or indirect effects of effort and skill acquisition [102] (Gevensleben, Albrecht, et al., 2014). However, the outcomes seem to be mixed, as fewer than 70% of those treated with NF improve self-regulation [94] (Aggensteiner et al., 2019) and only about 50% show the expected “dose-response” relation between learned regulation and clinical improvement [103] (Drechsler et al., 2007). Specifically, three studies found some significant association between brain self-regulation and ADHD core symptoms after SCP-NF [94,103,104] (Aggensteiner et al., 2019; Drechsler et al., 2007; Strehl et al., 2006). However, some recent frequency-band NF studies could not find any association between self-regulation and symptom reduction after treatment end [36,105] (Arnold et al., 2020; Janssen et al., 2016) or were contrary to the expectations, with associations found in the semi-active control group [94] (Aggensteiner et al., 2019). Interestingly, in the study from Arnold et al., 2020, significant association was found at 6-month follow-up, suggesting a possible specific delayed effect. These brain-behaviour association analyses are necessary to be able to disentangle specific from unspecific effects. However, so far, no firm general conclusion can be drawn regarding the specific effects related to self-regulation.

Predicting who responds to EEG-NF is particularly relevant. One SCP neurofeedback study found that increased theta activity predicts clinical responses to theta-modulating neurofeedback, and that stronger oscillatory parietal alpha activity along with stronger task-related preparatory SCPs together explained nearly 30% of the clinical outcome variance after SCP-NF [106,107,108] (Gevensleben, Kleemeyer, et al., 2014; Gevensleben, Moll, et al., 2014; Wangler et al., 2011). However, these intriguing results await independent replication.

Future studies should systematically investigate the specificity of self-regulation and the mechanisms which underlie the individual clinical effects, considering also reduced medication use, and long-term improvement in ecological settings. Also, whether individualisation of NF (e.g., limiting TBR training to those with elevated TBR) improves outcomes remains to be tested with appropriate control conditions.

#### 4.1.2. fMRI-Neurofeedback

Real-time fMRI neurofeedback (fMRI-NF), despite its lower temporal resolution relative to EEG-NF (seconds compared to milliseconds), has superior spatial resolution (millimetre rather than centimetre) and has the advantage that it can target the key cortical and subcortical brain function deficits that have been established in ADHD over the past 25 years of fMRI research [51] (Rubia, 2018). fMRI-NF enables participants to self-regulate the blood-oxygen level-dependent (BOLD) response of a targeted brain region or network through real-time feedback of their brain activity and has shown some promise in improving clinical symptoms and cognition in psychiatric disorders [109] (Thibault, MacPherson et al., 2018). To date, however, there are only two published fMRI-NF studies in ADHD.

The first fMRI-NF study was a small underpowered randomised controlled trial in seven adults with ADHD who underwent four weekly 1-h fMRI-NF of dorsal anterior cingulate cortex (dACC), combined with a mental calculation task, while six ADHD patients completed the same task in the scanner but were presented with visual cues indicating level of task difficulty instead of fMRI-NF [110] (Zilverstand et al., 2017). Both groups significantly increased dACC activation over the NF runs, including the transfer runs, and improved in an interference inhibition task. Both groups showed trend-level improvements in ADHD symptoms but did not differ from each other. However, only the neurofeedback group showed significantly stronger performance improvement in sustained attention and working memory tasks after treatment, but not the ADHD group that received no fMRI-NF, indicative of some positive effects of fMRI-NF of dACC on cognition in adults with ADHD [110] (Zilverstand et al., 2017).

A randomised controlled trial from our lab tested fMRI-NF of the rIFC compared to fMRI-NF of the left parahippocampal gyrus (lPHG) in adolescents with ADHD [111] (Alegria et al., 2017). Thirty-one boys with a clinical ADHD diagnosis underwent 11 runs of 8.5 min of fMRI-NF during 4 h-longs scans over a 2-week period, with a rocket movie as feedback. Eighteen participants learned to self-upregulate the target region, the rIFC (rIFC-NF group), while 13 participants self-upregulated a control region, the lPHG (lPHG-NF group). In both groups, activation of their target regions increased linearly across the 11 fMRI-NF runs. However, only the rIFC-NF group showed a transfer effect (self-regulation without feedback, as a proxy of transfer to real life) that significantly correlated with reduced ADHD symptoms. Although ADHD symptoms significantly improved in both groups, only the rIFC-NF group showed a large reduction of symptoms at 11 months follow-up, with an effect size of almost 1, compared to a trend-level reduction in the lPHG-NF group. Only the rIFC-NF group also showed trend-level improvement in their sustained attention performance [111]. In addition to the linear increase of activation of the rIFC in the rIFC-NF group, there was an increase in functional connectivity between the rIFC and the ACC and caudate, and a decrease in functional connectivity between the rIFC and regions of the posterior default mode network (DMN). This suggested that the NF of an isolated region led to positive network changes in cognitive control and DMN networks (Rubia et al., 2019) [65].

In order to measure the effects of fMRI-NF on brain function in ADHD, the participants of this study also performed a motor response inhibition fMRI task, the tracking stop signal task, before and after fMRI-NF. There was a significant group by time effect for the fMRI data, where post minus pre fMRI-NF, the rIFC-NF relative to the lPHG-NF group, had increased activation during successful inhibition in the rIFC and parietal regions [111] (Alegria et al., 2017) and increased activation in left-hemispheric error monitoring regions of IFC, premotor cortex, insula and putamen during failed inhibition, which furthermore correlated with ADHD symptom improvements and were concomitant with increased post-error reaction time adjustment at the behavioural level [112] (Criaud et al., 2020). Interestingly, we observed similar upregulation effects in ADHD children in the same regions when comparing the effects of stimulant medication relative to placebo using the same stop task [14,113,114] (Cubillo et al., 2014; Rubia et al., 2014; Rubia et al., 2011), suggesting that fMRI-NF of the rIFC has similar brain activation effects on the disorder as stimulant medication, but without the side effects. In fact, we found no group differences in side or adverse effects after fMRI-NF.

However, not everyone is capable of learning fMRI-NF. Similar to the EEG-NF literature [101,115], we found that only 48% of patients learned successfully to upregulate their target region with fMRI-NF [116] (Lam et al., 2020). Furthermore, fMRI-NF learning was better predicted by fMRI than clinical or cognitive data. Thus, increased activation in left inferior fronto-striatal cognitive control regions and reduced activation in posterior temporo-occipital and cerebellar regions during successful inhibitory control in the fMRI stop task predicted fMRI-NF self-regulation capacity. Clinical measures were not associated with general fMRI-NF learning and within a task battery of executive function tasks, only faster processing speed during inhibition and attention tasks predicted fMRI-NF learning (Lam et al., 2020) [116].

#### 4.1.3. NIRS Neurofeedback

Only one pilot study so far tested the related neural haemodynamic modulation method of NIRS Neurofeedback (NIRS-NF) of the left dorsolateral prefrontal cortex in nine ADHD children, compared to EEG-NF (*n* = 9) and electromyography-NF (*n* = 9). Only NIRS-NF resulted in significant improvements in clinical ADHD symptoms and in cognitive inhibition and attention functions after 11 hourly sessions over 4 weeks, which was, however, not superior to EEG-NF or electromyography-NF (Marx et al., 2015) [117].

#### 4.1.4. Conclusions from Neurofeedback Studies

In conclusion, EEG-NF has been tested in ADHD for about 45 years, and a large number of meta-analyses of randomised controlled trials of EEG-NF show consistent small to medium effect sizes for symptom improvements, but still with controversy regarding “probably” blinded raters [32,80] (Bussalb et al., 2019; Cortese et al., 2016). Furthermore, the specific effects of EEG-NF and the association between NF self-regulation and clinical improvement are still unclear and need more systematic research. Additionally, future studies should optimize the designs to promote EEG-NF self-regulation and improvement over time, considering increased artefacts and altered reward learning in ADHD [118] (e.g., Aase & Sagvolden, 2005) and further systematically investigate why some participants show low regulation performance.

fMRI-NF and NIRS-NF research is still in its infancy. Some of the findings of these small proof-of-concept studies using fMRI-NF and NIRS-NF are promising. However, larger, double-blind, placebo-controlled randomised controlled trials need to further assess the potential efficacy of fMRI or NIRS-NF in ADHD. In rtfMRI-NF or NIRS-NF, nothing is known on optimal protocol, such as the optimal regional target of neurofeedback, number and duration of neurofeedback sessions, whether there is potentially a saturation or a plateau of self-regulation in specific brain regions, and after how many sessions, or how and which interindividual differences affect learning of brain self-regulation. Moreover, transfer effects on clinical behaviour are unclear. Other untested questions are optimal reinforcement strategies or cognitive strategies when applying fMRI or NIRS-NF in children. In addition, the positive or negative side effects of regional fMRI-NF on not self-regulated regions or on non-targeted cognitive functions have never been tested in neurofeedback studies. It is entirely possible that the self-regulation training of a particular brain region has a downregulation effect on neighbouring, interconnected or contralateral regions and the potential costs of such downregulations need to be assessed. In fact, our fMRI-NF study in adolescents with ADHD, for example, showed a reduction in the active rIFC group in activation of the parahippocampal control region, while the control group had a decrease in right IFC activation, suggesting that the self-regulation of a particular region leads to the downregulation of other regions in the brain (Alegria et al., 2017) [111].

One of the most interesting findings from the existing NF studies is evidence for longer-term delayed consolidation effects which appear to be more pronounced at follow-up than at post-NF treatment assessment points [76,111,117,119] (Alegria et al., 2017; Arns et al., 2014; Arns & Strehl, 2013; Marx et al., 2015); however, one recent study showed no superiority over a semi-active control group at six months follow-up [94] (Aggensteiner et al., 2019). Such delayed consolidation effects reinforce the notion that brain self-regulation via NF affects neuroplasticity and may hence have unique longer-term efficacy. This would be a clear advantage over pharmacological medication such as stimulants which do not affect neuroplasticity and may even lose efficacy over time [13,120] (Cortese et al., 2018; Molina et al., 2009) due to potential brain adaptation [121] (Fusar-Poli et al., 2012). In fact, neuroplasticity of neurofeedback has been demonstrated in humans in the form of cortical excitability changes, white matter tract and structural changes [122] (Sitaram et al., 2017). The stability of these changes over time, is, however, unknown. This potential for longer-lasting neuroplastic effects and the apparent lack of side effects are likely to be the main attraction of neurofeedback therapies.

### 4.2. Brain Stimulation

Non-invasive brain stimulation therapies, most prominently rTMS and tDCS, have only over the past decade been applied to ADHD. These stimulation techniques affect cellular and molecular mechanisms involved in use-dependent local and distant synaptic plasticity, i.e., GABA and glutamate-mediated long-term potentiation, which may lead to longer-term effects [123] (Demirtas-Tatlidede et al., 2013). In fact, several studies in healthy populations and patient groups have shown longer-term cognitive effects of up to 1 year after stimulation [124,125] (Ruf et al., 2017, Katz et al., 2017). With high relevance to ADHD, positron emission tomography (PET) studies have shown that anodal frontal tDCS can release neurotransmitters such as dopamine [126,127,128] (Fonteneau et al., 2018; Meyer et al., 2019 Borwick et al., 2020), which furthermore correlated with better attention [129] (Fukai et al., 2019), with some indirect evidence for effects on noradrenaline [130,131] (Adelhöfer et al., 2019; Mishima et al., 2019). Similarly, rTMS over prefrontal regions in animals and humans has been shown to induce changes to neurotransmitter systems including alterations to serotonin, striatal dopamine release and metabolite levels, as well as to the release and concentrations of striatal glutamate [132,133] (Moretti et al., 2020; Poh et al., 2019). It has furthermore been shown that the combination with cognitive training which can prime the areas to be stimulated with a cognitive task is more effective than stimulation alone, due to the synergistic effects of functional targeting (Cramer et al., 2011; Kuo & Nitsche, 2012; Ziemann & Siebner, 2008) [132,133,134,135,136].

#### 4.2.1. Repetitive Transcranial Magnetic Stimulation (rTMS)

rTMS is a non-invasive and relatively safe brain stimulation technique that uses brief, intense pulses of electric current delivered to a coil placed on the subject’s head in order to generate an electric field in the brain via electromagnetic induction. A commonly used figure-8 coil can provide relatively focal stimulation of approximately 5 mm^3^. The induced electrical current triggers action potentials in the brain via current flowing parallel to the surface of the coil and thus modulates the neural transmembrane potentials and therefore neural activity. The magnitude of the stimulation is inversely related to the distance from the coil. The effect differs depending on the stimulation intensity and duration; number of stimulation pulses and their frequency per second; and coil orientation. In general, based on motor studies, high frequency (>5 Hz) rTMS promotes cortical excitability, while low frequency (1 Hz) rTMS inhibits cortical excitability [137] (Lefaucheur et al., 2014).

Longer-term clinical improvements with rTMS have been demonstrated in several psychiatric disorders [138,139] (Janicak & Dokucu, 2015; Mehta et al., 2019), supporting its neuroplastic potential. Relative to tDCS, rTMS has greater specificity in targeting neural regions [140] (Parkin et al., 2015), but is more expensive. The most common side effects are transient scalp discomfort underneath the coil due to stimulation of the pericranial muscles and peripheral nerves [141] (Rossi et al., 2009).

The majority (four out of six) of rTMS studies were conducted in adults with ADHD. Two double-blind, sham-controlled crossover studies targeted the right DLPFC. In 13 ADHD adults, one session of 20 Hz-rTMS relative to sham significantly improved overall self-rated ADHD symptoms and inattention but had no effect on hyperactivity, mood or anxiety scores [142] (Bloch, 2012). In nine ADHD adults, 10 daily sessions of 10 Hz-rTMS relative to sham showed no effect on self-rated clinical symptoms, nor on EEG or EF measures [143] (Weaver et al., 2012). In a single-blind, sham-controlled, randomised study in 22 ADHD adolescents, 20 daily sessions over 4 weeks of 18 Hz deep rTMS over bilateral DLPFC (*n* = 13) compared to sham (*n* = 9) showed no effect on self-rated clinical or cognitive measures of sustained attention [144] (Paz et al., 2018). A parallel, semi-blind, randomised, active and sham-controlled study in 43 young adults with ADHD tested 15 sessions of 18 Hz-rTMS over 3 weeks and a 1-month follow-up maintenance session over the right prefrontal cortex, targeting both DLPFC and IFC. Stimulation was combined with a short cognitive training session targeting the right prefrontal cortex, which was conducted before and after stimulation. While patients were blind, researchers were only blind for the sham and real but not the active stimulation control condition, which was an off-target focal stimulation 5–6 cm away from the DLPFC or IFC and which did not target DLPFC or IFC [145] (Alyagon et al., 2020). The DLPFC/IFC stimulation compared to the other conditions showed significant improvements in the primary clinical outcome measure, which was self-rated ADHD symptoms, with an effect size of 0.96 versus sham and 0.68 versus the active control stimulation, and there was only a significant improvement in the hyperactivity/impulsiveness in the self-rated subscales. Superiority of real versus control conditions was no longer significant at follow-up a month later. There were no significant effects on depression ratings, behavioural executive functions (as measured on the BRIEF) or cognitive inhibition measures except for a trend of improving Stroop task performance relative to sham but not active control, which was correlated with the clinical changes in the DLPFC/IFC stimulation group. EEG measures showed a negative correlation between alpha activity and a positive correlation between low gamma activity under the stimulation area with clinical symptom improvements in the DLPFC/IFC stimulation group.

Two studies were conducted in children with ADHD. An open label tolerability and safety trial in 10 children with ADHD without a sham condition showed fewer teacher-rated inattention and parent-rated hyperactivity/impulsivity symptoms one week after five daily sessions of 1 Hz-rTMS over left DLPFC compared to baseline [146] (Gómez et al., 2014). A larger study randomised 60 children with ADHD into either 30 daily 25 min sessions of 10 Hz rTMS over right DLPFC, Atomoxetine (1.2 mg/kg) or combined treatment over 6 weeks. The combined treatment group compared to the individual treatment groups improved significantly post- relative to pre-treatment in inattention and hyperactivity/impulsiveness, but not in oppositional defiant behaviours, nor in cognitive measures of working memory, sustained attention and reward-based decision making. All groups improved in these clinical and cognitive measures [147] (Cao et al., 2018). However, without a sham condition, placebo or practice effects cannot be ruled out in both studies (Table 1).

With respect to safety, one study reported a seizure in one patient (who was excluded from the study) after three sessions [145] (Alyagon et al., 2020), while most other studies reported no or few side or adverse events other than transient headaches and scalp discomfort localised to the stimulation area.

In conclusion, rTMS is relatively safe. The majority of studies were conducted in relatively small samples, using few session numbers of rTMS, and two out of six studies did not include a sham condition, making it impossible to rule out placebo or practice effects. Based on the conducted studies so far, there is relatively little evidence that several sessions of rTMS improve ADHD symptoms or cognition, with the exception of one study in adults that used multisession rTMS and stimulated right DLPFC and IFC combined with cognitive training, which needs replication. More multisession sham-controlled RCTs in large patient numbers are needed, in particular in paediatric ADHD, to more thoroughly test TMS effects using different protocols.

#### 4.2.2. Transcranial Direct Current Stimulation (tDCS)

tDCS applies a weak continuous direct electric current to underlying brain regions via scalp electrodes with the electrical current passing between a positively charged anode and a negatively charged cathode. In general, currents induce plasticity by causing subthreshold polarity-dependent increases (anodal stimulation) or decreases (cathodal stimulation) in membrane potentials that modify spontaneous discharge rates and cortical excitability, thus increasing/decreasing cortical function and synaptic strength [72] (Ashkan et al., 2013). tDCS is much easier to apply and has lower financial costs than TMS. Furthermore, tDCS has the advantage of being less painful than TMS and hence is more suitable for paediatric applications. Side effects are minimal in children (and adults), typically involving transient itching and reddening of the scalp site of stimulation in some participants (Krishnan et al., 2015; Zewdie et al., 2020) [148,149].

Combining cognitive training with tDCS over a cortical area that mediates the cognitive function being trained [134] (Kuo & Nitsche, 2012) has been shown to yield larger and long-lasting functional improvements that modify the impaired system [136] (Cramer et al., 2011), presumably via a synergistic effect of training-induced and stimulation-induced plasticity [135] (Ziemann & Siebner, 2008). Combined effects of cognitive training with tDCS in other disorders and healthy subjects have been shown to last up to 6 months [150,151] (Boggio et al., 2012; Kuo et al., 2014) and 1 year (Katz et al., 2017) [125].

Functional neuroimaging studies have furthermore demonstrated modulation not only of the stimulation site but also of functionally interconnected (sub)cortical regions [152] (Polania et al., 2011), which makes them useful for targeting networks such as fronto-striatal systems in ADHD. Furthermore, relevant to ADHD, striatal dopamine [153] (Pogarell et al., 2007) and noradrenaline [131,154] (Kuo et al., 2017; Mishima et al., 2019) have been implicated in the mechanism of action, both of which are typically reduced in ADHD (Cortese et al., 2018) [13].

Unlike with rTMS, the majority of tDCS studies (12 out of 18) (see Table 2) have been conducted in children with ADHD, presumably due to the high tolerability and relatively low side effect profile of tDCS, which would make it a good treatment option if efficacious. The majority of studies used very small session numbers and tested cognitive effects only (see Table 2).

Two double-blind, sham-controlled, crossover studies applied single session stimulation over the DLPFC. In 15 adolescents with ADHD, anode-left/cathode-right tDCS over bilateral DLPFC improved WCST completion time, n-back reaction times and Stroop reaction times and commission errors to incongruent trials but had no effect on n-back accuracy or Go/No-Go task performance [155] (Nejati, Salehinejad et al., 2020). In 10 ADHD adolescents, anodal tDCS over the left dlPFC improved n-back accuracy and reaction times compared to both sham and cathodal tDCS; anodal and cathodal tDCS also improved WCST performance, but anodal tDCS led to greater improvement; cathodal tDCS also improved No-Go accuracy, potentially via interhemispheric inhibition increasing right prefrontal activation [155] (Nejati, Salehinejad et al., 2020), a region associated with motor response inhibition in children and adults [156,157,158] (Rubia et al., 2013; Rubia et al., 2003; Rubia et al., 2007). This last finding is in line with a single-blind, crossover study in 21 adolescents with ADHD, which found in a subsample of seven participants that, compared to sham, one session of anodal, but not cathodal, tDCS over the right IFC reduced commission errors (trend-level) and reaction time variability in an interference inhibition task (Breitling et al., 2016) [159].

Two single-blind, sham-controlled crossover studies stimulated left DLPFC or right IFC in 20 high school students with ADHD symptoms that were above cut-off on validated ADHD questionnaires. Single session anodal relative to cathodal tDCS over the left DLPFC improved Go accuracy while cathodal tDCS relative to anodal tDCS and sham improved No-Go accuracy in the Go/No-Go task, but there was no change in Stroop task performance [160] (Soltaninejad et al., 2019). Anodal tDCS over the rIFC relative to sham improved Go accuracy, but there were no changes in other Go/No-Go or Stroop task measures (Soltaninejad et al., 2015) [161].

A double-blind, sham-controlled RCT in 50 children with ADHD tested the effects of 15 sessions of 20 min of right IFC anodal tDCS combined with cognitive training in executive function tasks. The study found that both groups improved in clinical symptoms and cognitive functions, but the improvement in the real versus sham tDCS in primary and secondary clinical outcome measures was significantly less pronounced [162] (Westwood et al., 2021). Groups did not differ in a large battery of executive function cognitive outcome measures [162] (Westwood et al., 2021) nor in EEG measures within a smaller subsample of data collected from 26 participants only [163] (Westwood et al., 2021). Furthermore, the real tDCS group had worse adverse effects related to mood, sleep and appetite immediately after stimulation (Westwood et al., 2021) [163].

A double-blind crossover study applied five daily sessions of anodal or sham tDCS over left DLPFC in 15 adolescents with ADHD, but because of a carry-over and learning effects, only the first sessions were analysed, thus reducing the sample to seven to eight participants per condition [164] (Soff et al., 2017). Compared to sham, anodal tDCS improved parent-rated inattention and cognitive measures of attention (QbTest; which combines cognitive measures of hyperactivity, impulsiveness and inattention in a hybrid n-back/GNG task) one week but not immediately after the last stimulation session, while cognitive measures of hyperactivity on the QbTest were improved immediately after anodal tDCS and seven days later [164] (Soff et al., 2017). Analysis of 13 out of the 15 ADHD adolescents after a single session of anodal tDCS relative to sham showed reduced reaction time variability but increased errors on the QbTest, but this analysis included the carryover effect [165] (Sotnikova et al., 2017).

A double-blind, sham-controlled crossover study found that overnight slow-wave oscillatory anodal tDCS over left and right DLPFC, relative to sham, improved declarative memory in 12 ADHD children [166] (Prehn-Kristensen et al., 2014), reaction time and its intra-subject variability on Go trials in a Go/No-Go task in 14 ADHD children [167] (Munz et al., 2015) but had no effects on No-Go accuracy, alertness, digit-span or motor memory.

An open label trial in nine ADHD children found that five daily sessions of anodal tDCS to left DLPFC combined with a picture association cognitive training task reduced errors in attention (omission) and switch-task performance, but did not improve working memory, while parents, with one exception, reported improvements in some of their children’s behaviour [168] (Bandeira et al., 2016).

In a double-blind crossover study in 14 children and adolescents with ADHD, the right IFC was stimulated with either conventional tDCS, high definition tDCS (HD-tDCS) or sham while performing a working memory task with inhibitory elements which was repeated after stimulation as an outcome measure. HD-tDCS is a 4:1 small electrode array with one electrode encircled by four electrodes of the opposite polarity, which delivers a more spatially restricted and therefore focal stimulation that can reduce side effects from stimulating non-target brain regions. The study found that neither a single session of conventional anodal tDCS nor HD-tDCS over right IFC combined with working memory performance compared to sham had any effect on performance in the n-back task; however, ERP data from 10 participants in ADHD showed elevated N200 and P300 after the two tDCS conditions versus sham and a shift towards the values seen in a healthy control group (Breitling et al., 2020) [169].

One study applied one session of anodal tDCS over the right inferior (and some superior) parietal lobe in 17 ADHD children in a single-blind crossover study. In line with the role of inferior parietal lobe in orienting attention, anodal relative to sham tDCS improved performance in bottom-up orienting attention but deteriorated selective attention as measured in the Stroop interference reaction time and error effects and had no effect on alerting or top-down executive attention as measured in the shifting attention and Go/No-Go tasks (Salehinejad et al., 2020) [170].

One recent study tested effects of tDCS on reward-related decision making in ADHD [171] (Nejati, Sarraj Khorrami, et al., 2020). Twenty children with ADHD received tDCS in three separate sessions with either anodal tDCS over the left DLPFC and cathodal tDCS over right vmPFC; the reversed montage; or sham stimulation. Anodal tDCS over the right vmPFC, coupled with cathodal tDCS over the left DLPFC, reduced risky decision-making in the Balloon Analogue R Task but had no effect on the key impulsiveness outcome measure in the delay discounting task (k mean); it did have an effect on some conditions, but these were not corrected for multiple testing (Nejati, Sarraj Khorrami et al., 2020) [171]. 

Another recent study compared the clinical and cognitive effects of tDCS with tRNS in ADHD. Although similar to tDCS in terms of equipment and setup, tRNS applies an alternating current at random frequencies and/or intensities. The mechanisms by which tRNS influences brain activity are less known but are thought to be different than for tDCS [172] (Fertonani & Miniussi, 2017). The most prevalent explanation for tRNS is stochastic resonance, whereby the introduction of an appropriate level of random noise enhances the output of subthreshold signals; thus, the application of weak electric currents amounts to an introduction of neural noise [172] (Fertonani & Miniussi, 2017). Information processing at the neuronal level is sensitive to stochastic resonance [173] (McDonnell & Ward, 2011). The double-blind cross-over study compared five sessions of transcranial random noise stimulation (tRNS) over left DLPFC and right IFC with tDCS of left DLPFC combined with executive function training in 19 children with ADHD. Relative to tDCS, tRNS showed a clinical improvement in ADHD rating scale scores from baseline after treatment and one week later. Cognitively, tRNS compared to tDCS improved working memory, but only processing speed during sustained attention. An exploratory moderation analysis predicted a trend-level larger tRNS effect on the ADHD rating scale for those patients who showed the greatest improvement in working memory. tRNS yielded fewer reports of side effects, in line with the literature on adults showing that tRNS is a more comfortable neurostimulation method than tDCS (Berger et al., 2021) [174].

Only four studies have been conducted in adults with ADHD. In a double-blind parallel study in 60 adults, anodal tDCS over the left DLPFC compared to sham had no effect in two Go/No-Go tasks or a functional cortical network activity based on EEG recordings in a subsample of 50 patients [175] (Cosmo et al., 2015). One single-blind crossover study applied a single session of anodal tDCS over the left and right DLPFC in 20 undergraduate students with ADHD, which, compared to sham, improved in hyperactivity measures (i.e., multiple/random responses) in a sustained attention task but had no effect on omission errors or reaction times [176] (Jacoby et al., 2018). A double-blind crossover study in 37 adults with ADHD administered three sessions of visual working memory training combined with anodal tDCS of the left DLPFC and reported that compared to sham, anodal tDCS reduced commission errors in a sustained attention task immediately but not three days after the last stimulation, while there was no effect on omission errors, reaction times, stop task or visual working memory training performance [177] (Allenby et al., 2018). [177] One double-blind parallel study in 17 adults with ADHD found that tDCS of anodal right/cathode-left DLPFC (*n* = 9) versus sham (*n* = 8) improved inattention but not hyperactivity/impulsive symptoms immediately after five daily sessions of stimulation and at a 2-week follow-up, with total ADHD scores also improving at the 2-week follow-up, although group differences disappeared at the 4-week follow-up (Cachoeira et al., 2017) [178]. Finally, in a double-blind crossover study in 37 adults with ADHD, participants were asked to perform a Flanker (*n* = 18) or a stop task (*n* = 19) before and after receiving a single session of anodal tDCS over the left or right DLPFC relative to sham. In the Flanker task, left but not right DLPFC stimulation reduced reaction times on incongruent but not congruent trials compared to sham and right DLPFC stimulation. This was furthermore correlated with increased left and right P300 increase in EEG measures on incongruent trials after left and right DLPFC stimulation compared to sham, respectively and with reduced N200 amplitude after left compared to right DLPFC stimulation. In the stop task, there was no effect in inhibitory measures, but left DLPFC stimulation relative to sham increased Go reaction time, which was correlated with increased P200 amplitude during Go trials (Dubreuil-Vall et al., 2020) [179].

In conclusion, only 3 out of 17 tDCS studies tested clinical effects. Two studies found that tDCS of left DLPFC improved clinical inattention symptoms while one study foundthat tRNS compared to tDCS improved ADHD symptoms (see Table 2).

With respect to cognition, most studies found effects in the performance of some but not other tasks, with little consistency in findings between studies, and most studies did not correct for multiple testing (see Table 2). Two meta-analyses tested for consistent findings of tDCS on cognition in ADHD. A meta-analysis of 10 studies (201 children/adults with ADHD) found that one to five sessions of anodal tDCS over mainly left DLPFC significantly improved cognitive performance in inhibitory control measures (Hedges’ *g* = 0.12) and in n-back reaction times (*g* = 0.66) [180] (Salehinejad et al., 2019). However, effect sizes were small, and the meta-analysis likely overestimated statistical significance by not controlling for interdependency between measures and conflated inhibitory with non-inhibitory cognitive measures [181] (Westwood et al., 2021). Addressing these and other limitations, a larger meta-analysis of 12 tDCS studies (232 children/adults with ADHD) found that one to five sessions of anodal tDCS over mainly left DLPFC led to small, trend-level significant improvements in cognitive measures of inhibition (*g* = 0.21) and of processing speed (*g* = 0.14) but not of attention (*g* = 0.18) (Westwood et al., 2021) [181].

In conclusion, the findings of the use of tDCS to improve ADHD symptoms and cognition have been mixed, with some positive results on improving cognition, with, however, very small effects sizes observed in meta-analyses (see also Table 1). However, the comparability of results was hampered by the large heterogeneity in study designs, stimulation parameters and site of anodal and cathodal stimulation. Larger samples and more homogeneously designed studies using a larger number of sessions of localised tDCS with and without cognitive training are needed to more confidently assess clinical and cognitive benefits.

Importantly, for both TMS and tDCS but also tRNS or tACS, systematic testing is needed to identify the optimal stimulation parameters that can elicit reliable clinical or cognitive effects. Parameters that should be tested include optimal stimulation sites, frequency, duration, and superiority of stimulation effects combined with cognitive training. For tDCS, tRNS and tACS, studies should consider if effects depend on age, electrode size and inter-electrode distance, the focality of stimulation and antagonistic effects of cathodal stimulation on the desired effect of the anodal stimulation. Children, for example, have thinner skulls and less corticospinal fluid, which means potentiation of the effects of brain stimulation compared to adults and optimal dosages cannot be easily transferred from adult studies. For example, cathodal tDCS at 1 mA, which has excitability-diminishing effects in adults, has shown to have excitatory effects in children and adolescents when applied over the motor cortex [182] (Moliadze et al., 2015). Stronger intensity might be needed for deeper regions, such as IFC, as opposed to more superficial regions, such as DLPFC, which might explain the null findings in studies of stimulation of rIFC in ADHD (Salehinejad et al., 2020). Clear and evidenced dosage guidance is therefore paramount for paediatric studies, especially since stimulation intensity and duration are non-linear [183] (Lefaucheur et al., 2017) and neuroplasticity changes are strongest during childhood development [184] (Knudsen, 2004). Furthermore, hardly anything is known on the longer-term efficacy of tDCS/tRNS/tACS or TMS protocols in ADHD. In healthy volunteers, up to 1-year longer-term cognitive effects have been observed with tDCS combined with cognitive training [125] (Katz et al., 2017) and up to 1 month in other psychiatric disorders [185,186] (Kekic et al., 2016; Moffa et al., 2018) with evidence for longer-term effects also with TMS in other psychiatric disorders (Janicak & Dokucu, 2015; Mehta et al., 2019) [138,139].

Given that tDCS is thought to affect neuroplasticity [187,188] (Kim et al., 2014; Nitsche et al., 2008), potential longer-term efficacy could be the real advantage of tDCS over stimulant medication. There is furthermore potential to combine tDCS with pharmacological or non-pharmacological treatments, in particular with cognitive training, as mentioned above.

While direct side effects appear to be minor and transient for non-invasive brain stimulation, such as itching and tingling over the stimulation site [148,170] (Krishnan et al., 2015; Salehinejad et al., 2020), there are, however, important neuroethical concerns about potential unknown negative effects of localised brain stimulation on the still-developing brain. It has been suggested that the neural state at the time of stimulation (Silvanto et al., 2008) [189] or baseline cortical excitation-inhibition levels may influence stimulation effects [190] (Krause et al., 2013), with those with suboptimal basal neural states likely to benefit more than those who already have an optimal activation pattern. This would suggest that application in psychiatric patient groups like ADHD who have suboptimal activation patterns may be more justified than its application as cognitive enhancer in healthy children and adults [191] (Cohen-Kadosh et al., 2012). It is also possible that the stimulation of a particular region negatively affects the activation in other regions, which could then impair non-targeted functions.

Inter-individual differences in traits, which may be associated with differences in baseline neural states, have in fact shown to affect the benefits or costs of brain stimulation. For example, subjects with high mathematical anxiety benefited in their reaction time to mathematical tasks with tDCS over DLPFC, while those with low mathematical anxiety had an impairment in reaction time. Moreover, both groups became worse in an interference inhibition task [192] (Sarkar et al., 2014), which could possibly reflect a negative effect of tDCS of DLPFC on the neighbouring IFC region, which mediates interference inhibition. Similarly, prefrontal stimulation improved automaticity of learning but impaired numerical learning mediated by parietal regions, while parietal stimulation impaired automaticity of learning mediated by prefrontal regions and improved numerical learning [193] (Iuculano & Kadosh, 2013). Inter-individual differences in brain activation at baseline are hence likely to determine whether patients benefit or not from tDCS over a particular brain region, suggesting that future brain stimulation treatment should be individualised based on baseline brain and cognitive dysfunctions. This is particularly pertinent given that there is heterogeneity in cognitive dysfunction in ADHD (Nigg et al., 2005; Roberts et al., 2017) [11,12].

Findings of cognitive costs of tDCS on functions mediated by other brain regions are particularly worrying in paediatric applications where the brain is still developing and more plastic. It will therefore be crucial to assess potential costs on non-targeted cognitive functions which may occur via indirect down-stimulation of other brain regions that are interconnected with the stimulated site and that mediate these non-*targeted* functions. This knowledge will be crucial to understand the risk-benefit ratio of localised brain stimulation to the individual patient and to children in particular. These worries of effects on non-targeted brain regions also apply to the neurofeedback studies. Most ethical considerations have concluded that there are no ethical reasons against tDCS in children and adolescents who have a medical condition that is handicapping and where potential side effects can be outweighed by benefits, while use of tDCS as cognitive enhancer in healthy children and adolescents is not advised [191] (Cohen Kadosh et al., 2012). These benefits and risks, however, will still have to be established in ADHD as well as in other childhood disorders.

**Table 2 cells-10-02156-t002:** Clinical and cognitive effects of sham-controlled tDCS studies.

				Stimulation Protocol					Outcome Measures (Bold/Underlined = Improvement; Cursive = Impairment)	
Study	Design	*n*	Mean Age	Anode/Cathode	mA	Sessions	Timing ^a^	Duration (mins)	Clinical	Cognitive
***Children***										
† Bandeira et al., 2016 [168]	Open label	9	11	L DLPFC/R SOA	2	5	Online	28	**Patient Global Impression of Improvement**	Visual Attention Test (**OM**); NEPSY-II-inhibition (**Switch errors**); Digit Span; Corsi Cubes
Breitling et al., 2016 [169]	Single-blind, sham-controlled, randomised, crossover	21	14	R IFC/L Cheek	1	1	Online	20	n/t	Flanker (Incongruent trials: **COM** ^c,d^ & **RTV** ^c^) ^e^
				L Cheek/R IFC	1	1	Online	20	n/t	Flanker
Munz et al., 2015 [167]	Double-blind, sham-controlled, randomised, crossover	14	12	L DLPFC/R Cheek; R DLPFC/L Cheek	0.25	1	Offline	25 (5 on, 1 off)	n/t	Go/No-Go (**Go RT & RTV**); Motor memory; Alertness
Nejati et al., 2020, Exp 1 [171]	Double-blind, sham-controlled, randomised, crossover	15	10	L DLPFC/R DLPFC	1	1	Offline	15	n/t	Go/No-Go; N-back (Acc, **RT**); Stroop (Incongruent trials: **COM & RT**); WCST (**Completion time**)
Nejati et al., 2020, Exp 2 [171]	Double-blind, sham-controlled, randomised, crossover	10	9	L DLPFC/R SOA	1	1	Offline	15	n/t	Go/No-Go; N-back (**Acc** ^c^, **RT**) ^d^; WCST (**Total categories completed**, **total & pers errors**) ^d^
				R SOA/L DLPFC	1	1	Offline	15	n/t	Go/No-Go (No--Go acc) ^d^; N-back; WCST (**Total categories completed**, **total & pers errors** ^c^) ^d^
Prehn-Kristensen et al., 2014 [166]	Double-blind, sham-controlled, randomised, parallel	12	12	L DLPFC/R Cheek; R DLPFC/L Cheek	0.25	1	Offline	25 (5 on, 1 off)	n/t	Declarative Memory (**Acc**); Alertness; Digit Span
Soff et al., 2017 [164]	Double-blind, sham-controlled, randomised, crossover	15	14	L DLPFC/Vertex	1	5	Online	20	FBB-ADHD(**Inattention** ^f^) ^g,h^	QbTest (**Inattention** ^f^; **hyperactivity** ^i^) ^g,h^
Soltaninejad et al., 2019 [161]	Single-blind, sham-controlled, randomised, crossover	20	16	L DLPFC/R SOA	1.5	1	Online	15	n/t	Go/No-Go (**Go Acc**) ^c,d^; Stroop
				R SOA/L DLPFC	1.5	1	Online	15	n/t	Go/No-Go (**NoGo Acc**) ^c,j^; Stroop
‡ Soltaninejad et al., 2015 [161]	Single-blind, sham-controlled, randomised, crossover	20	16	rIFC/L SOA	1	1	Online	15	n/t	Go/No-Go (**Go Acc**); Stroop
Sotnikova et al., 2017 [165]	Double-blind, sham-controlled, randomised, crossover	13	14	L DLPFC/Vertex	1	1	Online	20	n/t	QbTest (**RT**, **RTV** ^k^, *OMs*, *Acc*) ^l^
Breitling et al., 2020 [169]	Double-blind, sham- and HD-tDCS controlled, randomised, crossover	ADHD: 15HC: 15	13 (10–16)	R IFC/L SOA	1	3 with CT	Online	20	n/t	WM task; ERPs **N200**; **P300**
Salehinejad et al., 2020 [170]	Single-blind, sham-controlled, randomised, cross-over	19	9 (8–12)		1	2	Online	23	n/t	**ANT** (orienting); GNG; SAT; *Stroop*
† Westwood et al., 2021 [162]	Double-blind, sham-controlled, randomised, parallel	50	14	R IFC/L SOA	1	15	Online	20	*ADHD-RS; Conners 3P*	GNG; Stop; Simon; WCST; CPT; MCT; time estimation; NIH WM; Verbal Fluency
Nejati et al., 2020 [171]	Double-blind, sham-controlled, randomised, cross-over	20	9	L DLPFC/R vmPFC R DLPFC/L vmPFC Sham	1	1	Online	20	n/t	**BART**; **CDDT** (**k20**, **k10**)
† Berger et al., 2021 [174]	Double-blind, active controlled, randomised, cross-over	19	7–12	L DLPFC (tDCS)/R SOA L DLPFC/R IFC (tRNS)	0.75	5	Online	5	n/t	ADHD-RS; Working & short-term memory, Moxo-CPT (all improved with tRNS vs. tDCS)
***Adults***										
† Allenby et al., 2018 [177]	Double-blind, sham-controlled, randomised, crossover	37	32	L DLPFC/R SOA	2	3	Online	20	n/t	Conners CPT (**COM** ^m^); Stop Task
Cachoeira et al., 2017 [178]	Double-blind, sham-controlled, randomised, parallel	A: 9 S: 8	A: 31 S: 34	R DLPFC/L DLPFC	2	5	Offline	20	ADHD Checklist (**Inattention**, **Total**) ^n^; SDS (**after tDCS**); **ADHD total score 2 weeks**	None
Cosmo et al., 2015 [175]	Double-blind, sham-controlled, randomised, parallel	A: 30 S: 30	A: 32 S: 33	LDLPFC/R DLPFC	1	1	Offline	20	n/t	Go/No-Go
Jacoby et al., 2018 [176]	Single-blind, sham-controlled, randomised, crossover	20	23	L&R DLPFC/Cerebellum	1.8	1	Offline	20	n/t	CPT (**multi-button presses**)
Dubreuil-Vall et al., 2020 [179]	Double-blind, sham-controlled, randomised, crossover	37	18–67	L DLPFC/R SOA R DLPFC/R SOA	2	1	Offline	30	n/t	Flanker (**incongruent RT**) *n* = 18; L P300; L *N200*. Stop (**go RTs**); L **P200**. *n* = 19 Flanker; Stop

Abbreviations: A, active; Acc, accuracy; ANT, attention networking task; BAART, Balloon analogue risk taking task; CDDT, chocolate delay discounting task; COMs, commission errors; Conners 3P, Conners-3 Parent Rating Scale; CPT, continuous performance task; DLPFC, dorsolateral prefrontal cortex; FBB-ADHD, parents’ version of a German adaptive Diagnostic checklist for ADHD; L, left; mA, milliamps; mins, minutes; n/t, not tested; OMs, omission errors; cM, contralateral mastoid relative the other electrode; SOA, contralateral supraorbital area relative the other electrode; IFC, inferior frontal cortex; MCT: Mackworth Clock Task; NIH-WM, NIH Toolbox List Sorting Working Memory Test; N200; negative ERP component; P300; positive ERP component; R, right; RT, reaction time; RTV, reaction time variability or standard deviation of reaction times; S, sham; SAT, switching attention task; SDS, Sheehan Disability Scale; SSRT, stop-signal reaction time; WCST, Wisconsin task-sorting task. ^a^ Timing refers to whether cognitive performance was during (online) or after (offline) stimulation; ^c^ Would likely not survive multiple comparison correction; ^d^ Comparisons between stimulation conditions based on post-hoc LSD tests, which do not correct for multiple comparisons; ^e^ Based on underpowered analysis focusing on the first session, with seven participants per condition; ^f^ Improvement only seen seven days after the fifth anodal tDCS session; ^g^ Did not survive correction for multiple comparisons; ^h^ Based on underpowered analysis focusing on the first five sessions, with seven/eight participants per condition; ^i^ Improvement seen immediately after the fifth anodal tDCS session and seven days later; ^j^ Significant in comparison to cathodal tDCS only; ^k^ Based on a crossover interaction. tDCS reduced RT and RTV in one out of four conditions (2-back tasks), but this did not survive correction for multiple comparisons; ^l^ Included carryover effect raised by Soff et al., (2017); ^m^ Significant only immediately after anodal tDCS, not significant three days later; ^n^ Inattention improved immediately after anodal tDCS and after two weeks, while total score improved only after two weeks. † combined stimulation with cognitive training; ‡ originally published written in Persian language but was translated for us by the lead author Dr Zahra Soltaninejad.

#### 4.2.3. Other Stimulation Methods

Only one study has tested random noise stimulation (tRNS) in children with ADHD compared to tDCS and is described above. No studies have been conducted in ADHD with other stimulation methods such as transcranial alternative current stimulation (tACS).

External trigeminal nerve stimulation (eTNS), also known as transcutaneous supraorbital nerve stimulation (tSNS), is another non-invasive intervention with minimal side effects. Small electrical currents are transmitted transcutaneously via a self-adhesive, supraorbital electrode to excite (trigger action potentials) on the supratrochlear and supraorbital branches of the ophthalmic nerve (V1) located under the skin of the forehead. The supraorbital nerve is a branch of the first trigeminal division. The trigeminal nerve has widespread connections to the brain, in particular the reticular activation system, locus coeruleus, brain stem, thalamic, frontal and cortical areas (Shiozawa et al., 2014) [194] as well as effects on dopamine and noradrenaline, all of which have effects on arousal and attention and have been implicated in ADHD [51] (Rubia, 2018). Two studies have tested the efficacy of eTNS in ADHD. An 8-week, open trial, pilot feasibility study in 21 children with ADHD between 7–14 years showed significant reduction in the investigator-completed ADHD-IV-Rating scale (ADHD-RS), both for the inattentive and hyperactive/impulsive subscales and the parent-completed Conners Global Index and the Clinical Global Impression-Improvement as well as a reduction in the parent-completed Behaviour Rating Inventory of Executive Function (BRIEF) that measures executive functions in daily life. Patients with ADHD also improved after treatment in scores of depression, but not of anxiety. Furthermore, they tested performance on working memory and attention network tasks and found improvements in reaction times to interference stimuli, indicating positive effects on selective attention and inhibitory control as well as a trend-level improvement in response variability that is considered a measure of arousal and attention. eTNS was well tolerated with few side effects such as eye twitch and headache that were transient (McGough et al., 2015) [195].

The second study from the same group was a blinded, sham-controlled pilot study of eTNS in 62 children with ADHD from 8–12 years old. The investigator-rated ADHD-RS total score was significantly reduced in the active relative to the sham group, as well as the inattentive and hyperactive/impulsive sub-scores and the Clinical Global Impression-Improvement scores. There was furthermore a trend-level differential improvement in the active group for anxiety but not for depression (McGough et al., 2019) [196]. There were no serious adverse events and relatively minor and transient side effects such as headache or fatigue. Quantitative electroencephalography (qEEG) data showed increased power in the active relative to sham group in right frontal midline and inferior frontal regions after compared to before treatment, which furthermore correlated with improvements in the ADHD-RS total score and the ADHD hyperactive-impulsive subscores, suggesting mediation of clinical effects (McGough et al., 2019) [196]. The qEEG findings are partly in line with animal and human imaging studies that show that eTNS stimulates the activation of cortical and subcortical structures such as thalamus, amygdala, locus coeruleus, reticular activation system, prefrontal regions, anterior cingulate and insula [197,198] (Aston-Jones & Cohen, 2005; Cook et al., 2014). An activation increase in cortical and subcortical regions in ADHD could be the underlying mechanism of action given consistent evidence from us and others of dysfunction in ADHD in fronto-striato-thalamic neural networks [51] (Rubia, 2018). It is hence plausible that eTNS improves ADHD symptoms and cognition by stimulating the activation of dysfunctional fronto-striato-thalamo-cortical systems. Based on evidence from this small, underpowered pilot study, eTNS is now the only brain stimulation technique that is FDA-approved for ADHD. More evidence is clearly needed to demonstrate the efficacy and effectiveness of eTNS for reducing ADHD symptoms and to define optimal protocols such as repetition frequency, duration of stimulation, etc., similar to the other neurotherapies, as well as to understand its currently unknown underlying mechanisms of action.

## 5. Overall Conclusions

With the exception of EEG-NF, neurotherapeutics is still in its infancy in the field of ADHD.

A large number of meta-analyses of randomised controlled trials of EEG-NF show consistent small to medium effect sizes for symptom improvements, but there is controversy regarding blinded raters [13,32] (Bussalb et al., 2019; Cortese et al., 2016). Further systematic research needs to focus on the specificity of the effects of EEG-NF as well as on longer-term efficacy. Investigating criteria predicting individual responses will be crucial for precision medicine.

Neurofeedback studies using higher spatially resolved neuroimaging techniques such as NIRS and fMRI have only recently been piloted in ADHD, being powered mostly to show feasibility. However, some emerging promising findings in relatively small subject numbers demand further testing. Larger, sham-controlled RCTs that allow the identification of predictors of learning are necessary to establish whether NIRS or fMRI neurofeedback training has potential as a treatment for some individuals with ADHD. Optimal neurofeedback protocols are not known for either NIRS or fMRI and need systematic testing. Potential negative effects on non-regulated brain regions have not been tested in any of the neurofeedback modalities but need to be understood for ethical reasons.

Several non-invasive brain stimulation studies with heterogeneous study designs have been conducted in relatively small groups of ADHD children and adults, most of them using TMS or tDCS in either one or five sessions targeting mostly DLPFC or IFC, based on the dysfunction findings in fMRI studies conducted in ADHD over the last two decades. Studies using TMS have not been promising so far. Meta-analyses of tDCS effects mostly over DLPFC show small effect sizes for improving cognition (Salehinejad et al., 2019; Westwood et al., 2021). Only three studies, including a study using tRNS, tested for clinical improvements, with inconsistent findings with respect to improvement of inattention. Larger, sham-controlled studies are needed to further test the efficacy of tDCS on clinical and cognitive functions and potential costs on non-targeted cognitive or behavioural functions.

In addition, like for fMRI and NIRS-NF, knowledge is needed on the optimal stimulation protocols for different age and patient subpopulations (i.e., stimulation site, intensity, duration, frequency, electrode size, inter-electrode distance, etc.). It is likely that brain stimulation combined with cognitive training has a larger potential to enhance brain plasticity in ADHD than brain stimulation alone. This will also require the development of good cognitive training tasks that target ADHD-relevant functions to be used in combination with brain stimulation techniques. Given its minimal side effects, tDCS or tRNS are promising tools for the treatment of childhood onset psychiatric disorders since they provide the opportunity to positively influence atypical brain development early and possibly longer-term [199] (Krause & Kadosh, 2013). This promise, however, needs to be tested systematically in large RCTs of different protocols. Furthermore, there is some worrying evidence for potential costs of localised brain stimulation on other non-targeted functions, and this needs to be thoroughly investigated before clinical application. tRNS and TNS have shown promising effects on improving ADHD symptoms in proof-of-concept studies but will need replication.

Financial cost effectiveness of the different neurotherapies will also need to be taken into consideration. TMS is considerably more expensive than tDCS or related brain stimulation devices (like tACS, tRNS), with the TMS devices costing over USD 50,000 as opposed to a few hundred to USD 20,000 for tDCS. rTMS also requires larger office space. Furthermore, many manufacturers of rTMS have a pay-per-use business model, which is very cost-ineffective. The administration of rTMS is also more expensive as it requires substantially more training for a clinician/technician than tDCS. Some TMS devices even require localisation via an MRI scan which can add to the costs. tDCS, on the other hand, is small, portable, user-friendly and can be bought commercially and be used at home without a therapist. Even if both techniques are administered clinically by trained staff, cost-effectiveness analyses for application to pain show that tDCS is more cost-effective than rTMS, with lower costs and higher efficacy for pain [200] (Zaghi et al., 2018). Given that both techniques show comparable, relatively small effects in improving ADHD symptoms or cognition, tDCS may hence be more cost-effective. In addition, considering that tDCS is less painful, has less side effects and is more comfortable to apply, it is also more feasible for application to young children. TNS may be the most cost-effective non-invasive brain stimulation treatment as it seems to have a higher efficacy than either rTMS and tDCS in improving ADHD symptoms and can be bought commercially and applied at home during sleep without the need for a therapist at a relatively small cost of about USD 1000 per device with additional electrode costs.

Capital costs for fMRI-neurofeedback are far more expensive than EEG-NF or NIRS-NF, with hourly MRI scans costing typically between USD800–1000 and a device cost of several millions. Certified EEG-NF equipment costs under USD 10,000. However, so far evidence suggests that fMRI-NF learning can be achieved in relatively fewer sessions than EEG-NF [96] (Thibault et al., 2016). For ADHD, EEG-NF typically requires 25–40 sessions of 45–60 min (Arns et al., 2009), and full treatment costs for 30–40 sessions have been estimated at USD4000–6000, similar to pharmacotherapy over 5–10 years [201] (Garcia Pimenta et al., 2021). This means higher costs for the administering therapist as opposed to fMRI-NF, which requires fewer sessions. If fMRI-NF proves to be more effective than EEG-NF with a smaller number of sessions, then the higher scan costs could be partly offset against lower session and therapist time costs compared to EEG-NF, as qualified behavioural psychotherapeutic support during training may be important to consolidate and transfer clinical effects of NF. With respect to feasibility, however, there are few centres currently that have the necessary hardware and software to apply fMRI-NF or fNIRS-NF. EEG-NF is relatively more commonly offered at several private and some clinical centres in the world and has fewer exclusion criteria. In conclusion, the substantial knowledge acquired in cognitive neuroscience on ADHD has opened up to translational neuroscience studies in an attempt to use neurofunctional biomarkers as treatment targets for neurotherapeutics. Neurotherapeutics seem attractive for ADHD due to their safety and minimal or no side effects compared to medication treatments as well as their potential for longer-term neuroplastic effects, which drugs cannot offer. However, neurotherapies need to be more thoroughly tested for their short- and long-term efficacy, optimal “dose” effects (i.e., optimal target site; intensity of stimulation; frequency of stimulation/neurofeedback sessions), potential costs that may accompany the benefits and their potential for individualised treatment depending on clinical or cognitive ADHD subtypes. It is also likely that different clinical or cognitive subgroups of ADHD patients will benefit from either neurofeedback, brain stimulation or medication, and establishing this knowledge will be crucial to the benefit of individual patients.

## Figures and Tables

**Figure 1 cells-10-02156-f001:**
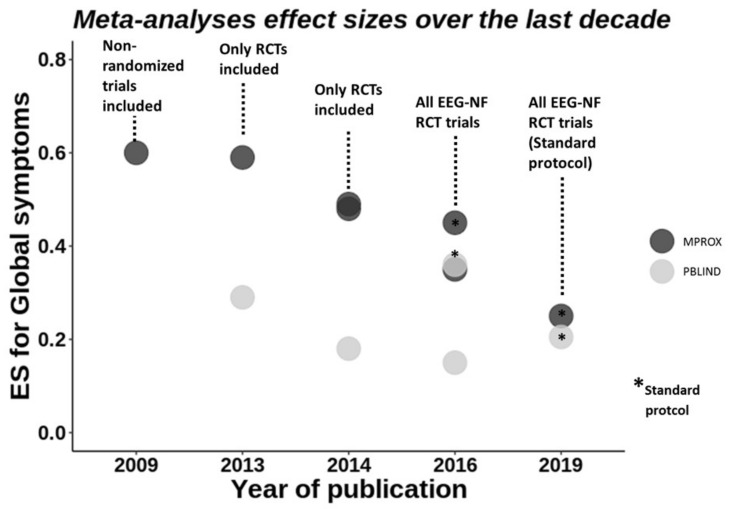
Effect sizes (ES) in meta-analyses of EEG neurofeedback studies for effects on global ADHD symptoms by year of publication. MPROX: ratings by parents/proximal raters; PBLIND: ratings by probably blinded raters. * Studies that used a standard protocol.

**Table 1 cells-10-02156-t001:** Clinical and cognitive effects of sham-controlled rTMS studies.

Stimulation Protocol								Outcome Measures (Bold/Underlined = Improvement)	
Study	Design	*N*	Age	Target	Sessions	Frequency	Duration	Clinical	Cognitive
***Children***									
Cao et al., 2020 [147]	Single-blind, randomised, parallel (2 active controls: ATX, ATX-rTMS; no sham)	64 (~20 each)	6–13	R DLPFC ^a^	20	18 Hz (100% MT)	2000 pulses (4 s on, 26 s off)	SNAP-IV	CPT; WISC; IGT
Gomez et al., 2014 [146]	Open label	10	7–12	L DLPFC	5	1 Hz (90% MT)	1500 pulses (on, off n/r)	DSM-IV ADHD symptom checklist (**hyperactivity**/**imp**., **inattention**)	n/t
***Adults***									
Bloch et al., 2010 [142]	Single-blind, sham-controlled, randomised, crossover	13	NR (adults)	R DLPFC ^a^	1	20 Hz (100% MT)	1680 pulses (2 s on, 30 s off)	PANAS (**inattention**, **total score**; mood, anxiety, hyperactivity); VAS (**inattention**, mood) ^b^	n/t
Paz et al., 2018 [144]	Double-blind, sham-controlled, randomised, parallel	A: 13 S: 9	A: 32 S: 30	L DLPFC ^c^	20	18 Hz (120% MT)	1980 pulses (2 s on, 20 s off)	CAARS	TOVA
Weaver et al., 2012 [143]	Single-blind, sham-controlled, randomised, crossover	9	18	R DLPFC ^a^	10	10 Hz (100% MT)	2000 pulses (4 s on, 26 s off)	CGI-I scale; ADHD-IV scale	WAIS/WISC-IV; Connors CPT; DKEFS; Buschke Selective Reminding Test; Symbol Digit Coding test; Finger Oscillation tasks
Alyagon et al., 2020 [145]	Double-semi-blind, randomised, active and sham-controlled	52 (15, 14, 14)	21–46	R IFC & DLPFC	15	18 Hz (120% MT)	1440 pulses (2 s on, 20 s off)	**CAARS** (global **ADHD symptoms**; **hyperactivity**/**impulsiveness**) (BAARS-IV (**hyperactivity**/**impulsiveness**), BRIEF-A, BDI)	STROOP; STOP

Abbreviations: A, active; BAARS, Barkley Adult ADHD Rating Scale; BRIEF-A, Behavioural Rating Inventory for Executive Functioning; BDI, Beck Depression Inventory; CAARS, Conners’ Adult ADHD Rating Scale; CGI-I, Clinical Global Impression-Improvement Scale; DKEFS, Delis–Kaplan Executive Function System; DLPFC, dorsolateral prefrontal cortex; Hz, number of magnetic pulses per second; IGT, Iowa Gambling task; L, left; MT, motor threshold; n/t, not tested; PANAS, Positive and Negative Affect Schedule; R, right; S, sham; SNAP-IV: Clinical rating scale of the severity of ADHD; TOVA, Test of Variables of Attention; VAS, Visual analogue scales; WAIS, Wechsler Abbreviated Scale of Intelligence, selected subtests from the Wechsler Adult Intelligence Scale; WISC-IV, Wechsler Intelligence Scale for Children-IV; ^a^ 5 cm forward to MT point; ^b^ small change from baseline of 0.25 and 1.16 out of 5-point Likert scales; ^c^ 6 cm rostral to motor cortex.

## Data Availability

Not applicable.

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
