# Peer review of "Neurotherapeutics for Attention Deficit/Hyperactivity Disorder (ADHD): A Review"

_cells, 2021, doi:10.3390/cells10082156_

Round 1

Reviewer 1 Report

This is narrative review about neurofeedback and non invasive brain stimulation interventions in ADHD over the lifespan. The topic is of high interest, the mansucript well written. I have only minor points/suggestions. 

  1. Even if it is a narrative and no systematic review, it would be beneficial to have a short methods section, in that the authors describe how literature research was done and which were the in- and exclusion criteria for the inlcuded studies and meta-analysis.
  2. Consideration of cost effectiveness might be included regarding the different neurotheraepeutics (for example MRI-neurofeedback vs.  EEG/NIRS neurofeedback, rTMS vs tDCS), taking costs and feasibility and the effect sizes into account. 
  3. Line 868 and following: instead of "cost-benefit" I suggest using "risk-benefit" and also in the following, "cost" is use whereas I would suggest to replace it with "risk" 

Author Response

  1. Even if it is a narrative and no systematic review, it would be beneficial to have a short methods section, in that the authors describe how literature research was done and which were the in- and exclusion criteria for the included studies and meta-analysis.

We now includes a new section 2 called “2. Search Methods of this review” after the 1. Introduction. Lines 91-104.

  1. Search methods for the review

For this review, Web of Knowledge, Scopus, PubMed, Ovid, Google Scholar, psyarxiv, and bioRxiv (up until the end of May 2021) were searched with the following key words: ‘electroencephalography’, ‘EEG’, ‘event-related potentials’, ‘ERP‘, ‘theta-beta ratio’, ‘TBR’, functional magnetic resonance imaging’, ‘fMRI’, ‘neurofeedback’, ‘EEG-neurofeedback’, ‘electrophysiology-neurofeedback’, ‘fMRI-neurofeedback’, ‘functional magnetic resonance imaging-neurofeedback’, ‘NIRS-Neurofeedback’, ‘near-infrared spectroscopy-neurofeedback’, ‘non-invasive brain stimulation’ ‘transcranial electric stimulation’, ‘transcranial direct current stimulation’, ‘tDCS’, ‘transcranial magnetic stimulation’, ‘repetitive transcranial magnetic stimulation’ ‘rTMS’,  ‘transcranial electric stimulation’, each in combination with ‘attention deficit/hyperactivity disorder’, ‘ADHD’, ‘hyperkinetic disorder’, inattention’, ‘hyperactivity’, ‘impulsivity’ and/or ‘meta-analysis’.  In addition, the reference lists of articles and reviews or meta-analyses were also hand searched. Single case studies were excluded.

  1. Consideration of cost effectiveness might be included regarding the different neurotherapeutics (for example MRI-neurofeedback vs. EEG/NIRS neurofeedback, rTMS vs tDCS), taking costs and feasibility and the effect sizes into account. 

Thank you. We have now added a section at the end of the general discussion on considerations of the cost effectiveness. Lines 1019-1054.

Financial cost effectiveness of the different neurotherapies will also need to be taken into consideration. TMS is considerably more expensive than tDCS or related brain stimulation devices (like tACS, tRNS), with the TMS devices costing over $50,000 as opposed to a few hundreds to $20,000 for tDCS. rTMS also requires larger office space. Furthermore, many manufacturers of rTMS have a pay-per-use business model, which is very cost-ineffective. The administration of rTMS is also more expensive as it requires substantial more training for a clinician/technician than tDCS. Some TMS devices even require localization via an MRI scan which can add to the costs. tDCS on the other hand is small, portable, user-friendly and can be bought commercially and be used at home without a therapist. Even if both techniques are administered clinically by trained staff, cost-effectiveness analyses for application to pain show that tDCS is more cost-effective than rTMS, with lower costs and higher efficacy for pain (Zaghi et al., 2018). Given that both techniques show comparable, relatively small effects in improving ADHD symptoms or cognition, tDCS may hence be more cost-effective. In addition, considering that tDCS is less painful, has less side effects and more comfortable to apply, it is also more feasible for their application to young children. TNS may be the most cost-effective non-invasive brain stimulation treatment, as it seems to have a higher efficacy than either rTMS and tDCS in improving ADHD symptoms and can be bought commercially and applied at home during sleep without the need for a therapist at a relatively small cost of about $1000 per device with additional electrode costs Capital costs for fMRI-Neurofeedback is far more expensive than EEG-NF or NIRS-NF with hourly MRI scans costing typically between $800-$1000 and a device cost of several millions. Certified EEG-NF equipment costs under $10,000. However, so far evidence suggests that fMRI-NF learning can be achieved in relatively fewer sessions than EEG-NF (Thibault et al., 2016). For ADHD, EEG-NF typically requires 25-40 sessions of 45-60 min (Arns et al., 2009), and full treatment costs for 30-40 sessions have been estimated at $4,000-$6,000, similar to pharmacotherapy over 5-10 years (Garcia Pimenta et al 2021). This means higher costs for the administering therapist as opposed to fMRI-NF, which requires fewer sessions. If fMRI-NF proves to be more effective than EEG-NF with smaller number of sessions, then the higher scan costs could be partly offset against lower session and therapist time costs compared to EEG-NF, as qualified behavioral psychotherapeutic support during training may be important to consolidate and transfer clinical effects of NF. With respect to feasibility, however, there are few centers currently that have the necessary hardware and software to apply fMRI-NF or fNIRS-NF. EEG-NF is relatively more commonly offered at several private and some clinical centers in the world, and has fewer exclusion criteria.

  1. Line 868 and following: instead of "cost-benefit" I suggest using "risk-benefit" and also in the following, "cost" is use whereas I would suggest to replace it with "risk" 

Thank you; this has now been replaced.

Reviewer 2 Report

This is a comprehensive review study on the neurotherapeutics for ADHD. Most of previous review studies focused on pharmacotherapeutic and psychotherapy for ADHD; the results of this study can provide the readers the deepened understanding of neurotherapeutics for ADHD.

  1. I would like to suggest the authors to make some minor revisions.
  2. According to DSM-5, the formal diagnosis is attention-deficit/hyperactivity disorder. The title warrants revision.
  3. The DSM-5 was published in 2013 but not 2000.
  4. Some paragraphs are too long to well follow-up. For example, 2.5. EEG-NF. The authors may consider to divide them into several shorter paragraphs for the readers to easily read.

Author Response

This is a comprehensive review study on the neurotherapeutics for ADHD. Most of previous review studies focused on pharmacotherapeutic and psychotherapy for ADHD; the results of this study can provide the readers the deepened understanding of neurotherapeutics for ADHD.

 Thank you.

  1. I would like to suggest the authors to make some minor revisions.
  2. According to DSM-5, the formal diagnosis is attention-deficit/hyperactivity disorder. The title warrants revision.

Thank you; this has now been changed in the title and the first sentence of the introduction.

3. The DSM-5 was published in 2013 but not 2000.

We apologize for the mistake; this has now been amended.

4. Some paragraphs are too long to well follow-up. For example, 2.5. EEG-NF. The authors may consider to divide them into several shorter paragraphs for the readers to easily read.

Thank you. We have now divided 2.5 into several shorter paragraphs and done this also for all other sections throughout the MS.